# Constitutive Model of Isotropic Magneto-Sensitive Rubber with Amplitude, Frequency, Magnetic and Temperature Dependence under a Continuum Mechanics Basis

**DOI:** 10.3390/polym13030472

**Published:** 2021-02-02

**Authors:** Bochao Wang, Leif Kari

**Affiliations:** 1CAS Key Laboratory of Mechanical Behavior and Design of Materials, CAS Center for Excellence in Complex System Mechanics, Department of Modern Mechanics, University of Science and Technology of China, Hefei 230027, China; 2The Marcus Wallenberg Laboratory for Sound and Vibration Research (MWL), Department of Engineering Mechanics, KTH Royal Institute of Technology, Teknikringen 8, 100 44 Stockholm, Sweden; leifkari@kth.se

**Keywords:** magneto-sensitive rubber, continuum mechanical frame, amplitude dependence, frequency dependence, magnetic dependence, temperature dependence

## Abstract

A three-dimensional nonlinear constitutive model of the amplitude, frequency, magnetic and temperature dependent mechanical property of isotropic magneto-sensitive (MS) rubber is developed. The main components of MS rubber are an elastomer matrix and magnetizable particles. When a magnetic field is applied, the modulus of MS rubber increases, which is known as the magnetic dependence of MS rubber. In addition to the magnetic dependence, there are frequency, amplitude and temperature dependencies of the dynamic modulus of MS rubber. A continuum mechanical framework-based constitutive model consisting of a fractional standard linear solid (SLS) element, an elastoplastic element and a magnetic stress term of MS rubber is developed to depict the mechanical behavior of MS rubber. The novelty is that the amplitude, frequency, magnetic and temperature dependent mechancial properties of MS rubber are integrated into a whole constitutive model under the continuum mechanics frame. Comparison between the simulation and measurement results shows that the fitting effect of the developed model is very good. Therefore, the constitutive model proposed enables the prediction of the mechanical properties of MS rubber under various operating conditions with a high accuracy, which will drive MS rubber’s application in engineering problems, especially in the area of MS rubber-based anti-vibration devices.

## 1. Introduction

Magneto-sensitive (MS) rubber is a kind of smart material whose main components are an elastomer matrix and magnetizable particles. The presence of these particles leads to a rapid increase in MS rubber dynamic modulus when a magnetic field is applied. This magnetic field-induced modulus increase is often referred to as the magnetic dependence of MS rubber. Due to the magnetic dependence, MS rubber-based devices can change their stiffness dynamically according to the response of the target structure and the external loading [1]. Therefore, a better anti-vibration effect can be achieved for MS rubber-based anti-vibration devices, such as dampers, absorbers, isolators and acoustic metamaterials, compared with the traditional rubber-based ones. For instance, a MS rubber-based damper was developed to reduce the rail noise and vibration [2]. The test results revealed that the effective frequency bandwidth of the MS rubber damper to reduce the noise and vibration can be broadened by changing the magnetic field applied. The effect of a MS rubber absorber to reduce the nonlinear vibration for a flexible arm was investigated and the simulation result reflected that the vibration energy transmitted by the flexible arm can be dissipated by altering the magnetic field applied on the MS rubber absorber [3]. With respect to the application of MS rubber in vibration isolators, a mathematical model for a cylindrical shaped MS rubber bushing standing on an infinite extended concrete floor was proposed by Blom and Kari [4] and the energy transmissibilities into the foundation with and without magnetic field were compared. The simulation result indicated that by changing the magnetic field at different frequencies, the energy flow into the foundation can be reduced. Following the path of Blom and Kari [4], Alberdi-Muniain et al. [5,6] measured the energy flow into the foundation by a MS rubber vibration isolator system experimentally and the same conclusion was obtained. Furthermore, for the anti-vibration effect of MS rubber isolators under random excitation, a fuzzy logic control algorithm was introduced by Wang and Kari [7] to control the magnetic field applied on a MS isolator. The result is that the anti-vibration effect of MS rubber isolator is better than the case where stiffness-changing is impossible for traditional rubber isolator. A similar investigation of the vibration control of MS rubber isolator under random loading can be found in Jung et al. [8]. Willey et al. [9] fabricated a reconfigurable MS rubber-based resonant acoustic metamaterial device where the addressable frequency range can be altered without remanufacturing. Harne et al. [10] combineed the ideas from topological controlled metamaterials and magnetic dependent modulus of MS rubber to create a MS rubber-based metamaterial device where the magnetoelastic metamaterial properties can be altered across orders of magnitude. Besides the application of MS rubber in vibration reduction area, due to the magneto-deformation ability of MS rubber, MS rubber shows a great potential in magnetic field-driven soft robot. Kim et al. [11] fabricated nanocomposite micro-actuators with grasping and walking function based on the programmable heterogeneous magnetic anisotropy method. Similarly, Lum et al. [12] and Hu et al. [13] developed magneto-elastic soft millimetre-scale robots with the function of swimming, climbing, rolling, walking and jumping. Qi et al. [14] used a 3D printing method to design the desired magnetic moment for the magneto-active soft materials and fabricated various biomimetric smart structures. Furthermore, by adding electrical conductive materials in MS rubber to increase the conductivity of MS rubber, an electrical sensitivity of MS rubber will exhibit in addition to the magneto sensitivity. For example, Bica et al. [15,16] fabricated a magnetoresistive sensor by adding graphene nanoparticles in MS rubber and then a theoretical model was proposed to explain the effect of the magnetic field intensity and pressure on the electrical conductivity of MS rubber. Similar research can be found in [17,18,19,20]. Due to the electromagnetic mechanical coupling effect, a great potential is shown for MS rubber in the area of vibration control, micro-electro-mechanical system and intelligent sensing and actuating.

In order to accelerate possible applications of MS rubber, a constitutive model which is able to predict the mechanical properties of MS rubber accurately is needed. Initially, Jolly et al. [21] derived the relationship between the magnetic dipole interaction stress and magnetization strength. The result is that the magnetic field induced modulus is proportional to the square of the magnetization strength. Based on the magnetic dipole-based model, the effect of the multi-chain, viscoelasticity and normal distribution of chains to the magnetic-induced shear modulus was taken into account by Zhu et al. [22], Chen et al. [23] and Yu et al. [24], respectively. The dynamic shear modulus measurement conducted by Blom and Kari [25] under different strain amplitudes and magnetic fields in a wide frequency range revealed that the shear modulus of MS rubber increases with increasing frequency while decreases with increasing strain amplitude. Therefore, modeling the frequency and amplitude dependence of MS rubber is also needed in additional to the magnetic dependence. The magnetic and frequency dependence of MS rubber can be depicted by the model developed by Kou et al. [26], Brancati et al. [27] and models based on the fractional derivative operator [28,29]. However, the amplitude dependence of MS rubber was not taken into account. After the observation of the amplitude dependence of MS rubber, several models has been proposed to simulate the amplitude dependence in the form of a smooth Coulomb friction model [30] and later with a bounding surface plastic model by Wang and Kari [31], respectively. Furthermore, Lejon et al. [32] considered the effect of prestrain on the mechanical behavior of MS rubber. However, for all the models mentioned, the constitutive equations are derived directly from the stress-strain law where a physically and thermodynamically consistent analysis is lacking.

Different from the magnetic-dipole based model, constitutive models of MS rubber based on the hyperelastic hypothesis and using the deformation gradient to describe kinematics were proposed by Dorfmann and Ogden [33,34], Kankanala and Triantafyllidis [35], Danas et al. [36] and Mukherjee et al. [37]. It is assumed that an augmented free energy, which is a function of the deformation gradient and magnetic flux density, exists within MS rubber. The stress and magnetic field intensity can be derived directly from the free energy function. Hence, the increase of internal energy by the magnetic field and the effect of the magnetic stress tensor on the mechanical behavior of MS rubber can be depicted. However, for the continuum mechanical-based model mentioned above, only the magneto-elastic behavior of MS rubber was considered. The inelastic behavior including frequency and amplitude dependence was neglected. Saxena et al. [38] introduced the magneto-viscoelasticity based on the model developed by Dorfmann and Ogden [33,34]. Wang and Kari [39] extended the model by introducing a bounding surface nonlinear kinematic hardening model to to simulate the amplitude dependency of MS rubber. Zhao et al. [40] used the continuum mechanical-based magneto-elasticity theory and developed a model for hard-magnetic particle based MS rubber.

Regarding the temperature dependence of MS rubber, measurement results revealed that there is a dramatic increase in the magnitude and loss factor of the dynamic shear modulus of MS rubber when the temperature decreases [41]. The temperature-dependent mechanical properties should be studied since in practical applications various temperatures may be encountered. Currently, there is a deficiency in the development of the temperature dependence model of MS rubber. While Zhang et al. [42] modeled the temperature-dependent mechanical properties of MS rubber by the Arrhenius function and Wan et al. [43] modeled the temperature dependence of anisotropic MS rubber by quadratic polynomial functions of shifting factors, the experiments form the frames of the models are all conducted above room temperature. To serve the application of MS rubber, modeling the temperature dependence of MS rubber below room temperature is needed as well. Turning away the attention from MS rubber to general rubber, normally, the William-Landel-Ferry (WLF) shift function [44,45,46,47] incorporated into a dashpot based generalized Maxwell model is utilized to model the temperature dependent viscoelastic behavior of rubber. However, the drawback is that a large number of Maxwell elements are needed to model the viscoelastic behavior of rubber accurately. Research results found that by utilizing a fractional standard linear solid (SLS) model combined with the WLF function, the temperature dependent viscoelasticity of rubber can be depicted more accurately with fewer parameters [48,49,50,51,52]. Therefore, a fractional SLS model with the WLF function is introduced to predict the temperature-dependent viscoelastic behavior of MS rubber in this paper. Parallelly, the Arrhenius function which uncovers the relationship between the temperature and reaction rate is widely used to model the temperature dependence of plasticity of crystalline solids [53,54]. Muhr [55] developed a model with the WLF function and the Arrhenius function together to capture the temperature, amplitude and frequency dependence of carbon black filler rubber. The simulation results show a good fit with the measurement results. Along the same line of Muhr’s [55] model, the temperature-related amplitude dependence of MS rubber will be modeled by the Arrhenius function for the elastoplastic part of the constitutive model developed in this paper. The constitutive model developed takes the magnetic, frequency, amplitude and temperature dependence into account, which covers the shortage that the amplitude and temperature dependence of MS rubber is not well modeled. The model developed in this paper will enhance the understanding and predicting of the mechanical behavior of MS rubber. More importantly, it will boost the application and design of MS rubber-based devices used in the sound and vibration area.

The structure of this paper is as follows. Fundamentals of continuum mechanics and magnetic equations is introduced in Section 2. Specific methods to model the magnetic, frequency, amplitude and temperature dependence is proposed in Section 3. The validation of the developed model through the comparison between experimental and simulation results is conducted in Section 4. Finally, a brief conclusion is drawn in Section 5.

## 2. Continuum Mechanics Frame and Magneto-Statics Basis

The derivation in this section mainly follows the work by Wang and Kari [39]. To guarantee the completeness of this paper, the fundamentals are introduced with clarity. Scalars are represented by italics and tensors (including first-order, second-order and forth-order tensors) are represented by bold-face letters. The rheological model is illustrated in Figure 1. A fractional SLS model is used to represent the viscoelastic behavior of MS rubber. The diamond shape in the fractional SLS model with parameters *a* and *b* represents the fractional derivative viscoelastic element. Parameters $G_{\infty }$ and $G_{\mathrm{ve}}$ represent the spring elements in the fractional SLS model. The elastoplastic element consisting of a spring $G_{\mathrm{ep}}$ in series with a plastic element with parameters $H_{p}$ and $S_{\mathrm{bounding}}$ is used to represent the amplitude dependence of MS rubber. A circle with $\mu_{0}$ and B is used to represent the magnetic stress tensor caused by the magnetic field and magnetic media interaction. The details related to the magnetic stress tensor will be introduced in Section 2.3.

### 2.1. Kinematics and Stresses

While the strain in the measurement of the mechanical performance of MS rubber which is related to the material parameter identication in the constitutive model in this paper is not large, a continuum approach is used to describe the motion of MS rubber in order that the work in this paper is consistent with the theory framework by Dorfmann and Ogden [33,34] where the quantities in the current and reference configurations are strictly distinguished. Assuming MS rubber is a continuum body, without loading and without magnetic field, MS rubber stays in the reference configuration $\Omega_{R}$ and the position of a typical point is X, as shown in Figure 2. After deformation from $\Omega_{R}$ to the current configuration $\Omega_{C}$, the new position is $x=\chi \left(X\right)$ and the deformation gradient is F=∂x∂x∂X∂X. The volume ratio $J\;=\;\det\left(F\right)$, which is the determinant of F, and the right Cauchy-Green strain tensor $C=F^{T}F$ are defined. Like many rubber material, MS rubber is regarded as an incompressible material. However, for the future finite element implementation need, a quasi-incompressible kinematic framework is applied. Therefore, the total deformation gradient is decoupled into a volume changing ($J^{1/3}I$) part and a volume preserving ($\bar{F}$) part by $F=\left(J^{1/3}I\right)\bar{F}$. Accordingly, the isochoric right Cauchy-Green strain $\bar{C}={\bar{F}}^{T}\bar{F}$, the isochoric Piola strain $\bar{e}=0.5\left({\bar{F}}^{-1}{\bar{F}}^{-T}-I\right)$ and the relative Piola strain ${\bar{e}}_{t}\left(s\right)=\bar{F}\left(t\right)\left[\bar{e}\left(s\right)-\bar{e}\left(t\right)\right]{\bar{F}}^{T}\left(t\right)$, which are used to obtain the viscoelastic stress are defined as well.

To represent the amplitude dependence, an intermediate configuration $\Omega_{I}$ is introduced to decompose the deformation gradient F into the elastic part $F^{e}$ and the plastic part $F^{p}$ for the elastoplastic element. Mathematically, it is
(1)$$F=F^{e}F^{p}.$$

The isochoric assumption of plasticity [56] follows, thus
(2)$$J=\det\left(F\right)=\det\left(F^{e}\right).$$

The isochroic elastic deformation gradient for the elastoplastic branch is
(3)$${\bar{F}}^{e}=J^{-1/3}F^{e}$$
and the corresponding isochoric elastic right Cauchy-Green strain is
(4)$${\bar{C}}^{e}={\left({\bar{F}}^{e}\right)}^{T}{\bar{F}}^{e}.$$

At the limit case of quasi-incompressibility, the spherical stress is determined by the boundary condition instead of the constitutive equations. Therefore, for the constitutive model developed herein, only the deviatoric stress is determined. Unless stated, otherwise, the stresses in this paper, are all deviatoric stresses. Corresponding to the rheological model in Figure 1, the total Cauchy stress σ consists of viscoelastic stress $\sigma^{\mathrm{ve}}$, elastoplastic stress $\sigma^{\mathrm{ep}}$ and magnetic stress $\sigma^{\mathrm{mag}}$,
(5)$$\sigma \;=\;\sigma^{\mathrm{ve}}+\sigma^{\mathrm{ep}}+\sigma^{\mathrm{mag}}.$$

By applying a pull-back operation [57], the corresponding second Piola-Kirchhoff viscoelastic stress $S^{\mathrm{ve}}$, elastoplastic stress $S^{\mathrm{ep}}$ and magnetic stress $S^{\mathrm{mag}}$ in the reference configuration can be obtained
(6)$$S^{\mathrm{ve}}=JF^{-1}\sigma^{\mathrm{ve}}F^{-T},$$
(7)$$S^{\mathrm{ep}}=JF^{-1}\sigma^{\mathrm{ep}}F^{-T}$$
and
(8)$$S^{\mathrm{mag}}=JF^{-1}\sigma^{\mathrm{mag}}F^{-T}.$$

Similarly, a pull-back operation of $\sigma^{\mathrm{ep}}$ to the intermediate configuration reaches the corresponding second Piola-Kirchhoff stress ${\hat{S}}^{\mathrm{ep}}$
(9)$${\hat{S}}^{\mathrm{ep}}=J{\left(F^{e}\right)}^{-1}\sigma^{\mathrm{ep}}{\left(F^{e}\right)}^{-T}$$
and the Mandel stress $\Sigma^{\mathrm{ep}}$ which is used to describe the elastoplastic behavior in the intermediate configuration
(10)$$\Sigma^{\mathrm{ep}}=C^{e}{\hat{S}}^{\mathrm{ep}}=J{\left(F^{e}\right)}^{T}\sigma^{\mathrm{ep}}{\left(F^{e}\right)}^{-T}.$$

### 2.2. Magnetic Field Equations

Before introducing the magnetic field equations, it should be noted that components $B_{R}$, $H_{R}$, $B_{C}$ and $H_{C}$ are all first order tensors. Quantities with subscript R and C belong to the reference and current configuration, respectively. According to Dorfmann and Ogden [33,34], MS rubber is assumed to be an electrically neutral insulation material where the effect of electric field on its mechanical performance can be neglected. The boundary conditions of the magnetic flux density $B_{R}$ and magnetic field intensity $H_{R}$ in the reference configuration are
(11)$$N\cdot ⟦B_{R}⟧=0$$
and
(12)$$N\times ⟦H_{R}⟧=0,$$
where ⟦·⟧ denotes the difference of quantities at the interface of two different magnetic media and N is the normal direction on the interface. Outside MS rubber, an electromagnet is utilized to generate the magnetic field. An isotropic magnetic media assumption is postulated and the permeability for metal is μ, then the relationship between $B_{R}$ and $H_{R}$ outside MS rubber is
(13)$$B_{R}=\mu H_{R}.$$

According to Dorfmann and Ogden [34], for any first-order Eulerian tensor T,
(14)$$\mathrm{Div}\left(T\right)=J\mathrm{div}\left(J^{-1}\mathrm{FT}\right)$$
and
(15)$$\mathrm{Curl}\left(T\right)=JF^{-1}\mathrm{curl}\left(F^{-T}T\right),$$
where $\mathrm{Div}\left(\cdot \right)$ and $\mathrm{Curl}\left(\cdot \right)$ are the divergence and the curl operator in the reference configuration while symbols $\mathrm{Div}\left(\cdot \right)$ and $\mathrm{Curl}\left(\cdot \right)$ are the corresponding divergence and curl operator in the current configuration. Along with the quasi-incompressible condition, the relationships between the magnetic field quantities in the reference and current configuration can be obtained
(16)$$B_{C}=J^{-1}FB_{R}$$
and
(17)$$H_{C}=F^{-T}H_{R}.$$

### 2.3. Augmented Stored Energy Function and Thermodynamic Analysis

Similar to Dorfmann and Ogden [33], an augmented stored energy Φ per unit volume in the reference configuration is defined
(18)$$\Phi =\Omega +0.5\mu_{0}^{-1}J^{-1}B_{R}\cdot \left(CB_{R}\right),$$
where Ω is the free energy per unit volume in the reference configuration for the mechanical stress $\sigma^{\mathrm{mec}}$ part, the term $0.5\mu_{0}^{-1}J^{-1}B_{R}\cdot \left(CB_{R}\right)$ represents the magnetic energy stored within MS rubber after applying the magnetic field and $\mu_{0}\;=4\pi \times 10^{-7}\;N\;A^{-2}$ is the vacuum magnetic permeability. Due to the correlation between the magnetic dependence and the amplitude dependence [25] along with the principle of material frame-indifference, the magnetic flux density in the reference configuration $B_{R}$ is incorporated into Ω. According to Wang and Kari [39], internal variable ζ is incorporated into Ω to account the irreversible process undergoes for the inelastic behaviors (frequency and amplitude dependency). Consequently,
(19)$$\Omega =\Omega \left(\bar{C},B_{R},\zeta \right).$$

It should be noted that the expression in Equation (Equation 19) reflects the whole mechanical part of the free energy. In the following part, Ω is decomposed into the viscoelastic and elastoplastic part. According to Saxena et al. [38], the corresponding Clausius–Planck inequality of thermodynamics during an isothermal process is
(20)$$-\dot{\Phi }+\frac{1}{2}S:\dot{C}+H_{R}\cdot {\dot{B}}_{R}⩾0.$$

By using the Coleman–Noll [58] procedure, it is obtained
(21)$$H_{R}=\frac{\partial \Omega }{\partial B_{R}}+\frac{1}{\mu_{0}J}CB_{R},$$
a dissipation condition
(22)$$-\frac{\partial \Omega }{\partial \zeta }\dot{\zeta }⩾0$$
and
(23)$$S=2\frac{\partial \Omega }{\partial C}+\frac{1}{\mu_{0}J}B_{R}⊗B_{R},$$
where ⊗ is the tensor product and $\mu_{0}^{-1}B_{R}⊗B_{R}$ reflects the anisotropy effect of the magnetic stress related to the magnetic field direction on MS rubber. By using Equations (Equation 6), (Equation 7) and (Equation 16), the equivalent of Equation (Equation 23) in the current configuration is
(24)$$\sigma =2J^{-1}F\frac{\partial \Omega }{\partial C}F^{T}+\frac{1}{\mu_{0}}B_{C}⊗B_{C}$$
with
(25)$$\sigma^{\mathrm{mec}}=\sigma^{\mathrm{ve}}+\sigma^{\mathrm{ep}}=2J^{-1}F\frac{\partial \Omega }{\partial C}F^{T}$$
and
(26)$$\sigma^{\mathrm{mag}}=\frac{1}{\mu_{0}}B_{C}⊗B_{C}.$$

## 3. Particularization of the Constitutive Model

### 3.1. Frequency Related Temperature Dependence

Normally, it is assumed that rubber material is thermo-rheological simple, then the time-temperature superposition principle can be applied to model the temperature dependency [59]. However, due to the imbed of particles, MS rubber may not show a thermorhelogical simplicity. Therefore, before modeling the temperature dependency, a rough check of the thermo-rheological simplicity of MS rubber is needed. The dynamic shear modulus measurement results of isotropic MS rubber with natural rubber as the rubber matrix under four kinds of magnetic field (0, 0.3, 0.55 and 0.8 T), three kinds of strain amplitude (0.0005, 0.0015 and 0.005) at discrete temperatures (242, 247, 254, 273, 293 and 306 K) ranging from 200 to 900 Hz conduced by Lejon and Kari [41] are utilized. The composition of the MS rubber is in Table 1. After the materials in Table 1 were mixed, 33% volume fraction of iron particles with size distribution of 77.7% 0–38 μm, 15.6% 38–45 μm and 6.7% 45–63 μm were stirred with the above mixture for approximately 5 min with a rotation speed of 30 rpm. Subsequently, the mixture was vulcanized for half an hour with a temperture of 150 °C under a conventional sulfur vulcanization system. Regarding the measurement setup and data analysis method, the reader is referred to Lejon and Kari [41].

According to Wan et al. [43], Rouleau et al. [60], as long as the Cole–Cole plot where the loss modulus versus the storage modulus in a logarithm coordinate for the measurement data lies close to one smooth curve, the thermo-rheological simplicity requirement is met. To perform this check, the results of the Cole–Cole plot for the measurement data of MS rubber at different strain amplitudes with 0 and 0.55 T magnetic field are presented in Figure 3, Figure 4 and Figure 5. It can be found that the measurement data at 242, 247, 254 and 273 K basically follows the same curve. While the results at 293 K follow the main curve sometimes, there is a certain deviation under different strain conditions. Furthermore, it can be found that the results at 306K of MS rubber does not follow the main curve. One possible explanation is that at these temperatures (293 and 306 K), the entropy elasticity of the rubber dominates. While the Cole–Cole plot reveals that the thermo-rheologicall-simplicity principle does not entirely hod for MS rubber, the studies by Guedes [61], Nakano [62], Çakmak et al. [63] show that the time-temperature superposition principle can still be exteneded to describe the temperature dependency of multi-component material as long as the deviations from the thermo-rheological simple behavior are not too large. Therefore, modeling of the temperature dependency by the time-temperature superposition principle of MS rubber is confined to measurement data at 242, 247, 254 and 273 K. The deviation of the test result at 242 K under 0.005 strain amplitude from the main curve can be tolerated since there is still a temperature-dependent amplitude dependence additional to the temperature-dependent viscoelasticity for the constitutive model.

The measurement results of the dynamic modulus of MS rubber from 242 to 273 K may cover the rubbery, transition and glassy region for MS rubber. Different from the magneto-elastic model by Dorfmann and Ogden [33,34] where only the equilibrium part of free energy exists, to model the frequency dependence, a fractional SLS is utilized. The free energy of the viscoelastic branch is
(27)$$\Omega^{\mathrm{ve}}=\Omega^{\mathrm{eq}}+\Omega^{\mathrm{ov}},$$
where $\Omega^{\mathrm{eq}}$ is the equilibrium part and $\Omega^{\mathrm{ov}}$ is the overstress part. For $\Omega^{\mathrm{eq}}$, a neo-Hookean quasi-incompressible hyperelastic model [64] is used with
(28)Ωeq=G∞I¯1−3I¯1−322,
where $\mathrm{tr}\left(\cdot \right)$ is the trace operator of second order tensors and ${\bar{I}}_{1}=\mathrm{tr}(\bar{C})$. Following the path of Wollscheid and Lion [65], the time-dependent free energy $\Omega^{\mathrm{ov}}$ is
(29)$$\Omega^{\mathrm{ov}}=-\int_{0}^{t}G\left(t-s\right)\frac{d}{ds}\mathrm{tr}\left[{\bar{e}}_{t}\left(s\right)\right]ds,$$
where G(t) is the relaxation function and *t* represents time. By applying the Coleman–Noll [58] procedure, the isochoric viscoelastic second Piola-Kirchhoff stress $S^{\mathrm{ve}}$ is
(30)$$S^{\mathrm{ve}}\left(t\right)=G_{\infty }J^{-2/3}\mathrm{Dev}\left(I\right)-\int_{0}^{t}G\left(t-s\right){\left[\det C\left(t\right)\right]}^{-1/3}P^{4}\left(t\right):\frac{d\bar{e}\left(s\right)}{ds}ds,$$
where $\mathrm{Dev}\left(\cdot \right)=\left(\cdot \right)-1/3\left[(\cdot ):C\right]C^{-1}$, $P^{4}\left(t\right)=\left[I^{4}-\frac{1}{3}C^{-1}\left(t\right)⊗C\left(t\right)\right]$ and $I^{4}$ is the forth order unit tensor. According to Wollscheid and Lion [65], the internal dissipation is
(31)$$D_{\mathrm{int}}=\int_{0}^{t}\frac{\partial^{2}G\left(t-s\right)}{\partial t^{2}}\mathrm{tr}\left[{\bar{e}}_{t}\left(s\right)\right]ds.$$

To satisfy the thermodynamic compatibility, it is required that $G^{\prime \prime }\left(t\right)\geq 0$ and $G^{\prime }\left(t\right)\leq 0$ where a prime denotes the derivative with respect to *t*. Regarding the detailed derivation of the fractional viscoelastic model under finite strain, the readers are referred to Haupt and Lion [66]; Lion et al. [67]; Wollscheid and Lion [65].

A Mittag–Leffler typed relaxation function is postulated
(32)$$G\left(t-s\right)=G_{\mathrm{ve}}E_{a}\left[-\frac{{\left(t-s\right)}^{a}G_{\mathrm{ve}}}{b}\right].$$

The symbol $E_{a}\left(\cdot \right)$ is the Mittag–Leffler function with the definition
(33)$$E_{a}\left(x\right)=\sum_{n=0}^{\infty }\frac{x^{n}}{\Gamma \left(1+na\right)},$$
where Γ represents the Gamma function and n∈N. It can be found that the thermodynamic compatibility can be satisfied by the relaxation function in Equation (Equation 32). After inserting Equation (Equation 32) into Equation (Equation 30), $S^{\mathrm{ve}}$ is
(34)$$\begin{array}{c} S^{\mathrm{ve}}\left(t\right)=G_{\infty }J^{-2/3}\mathrm{Dev}\left(I\right)-\int_{0}^{t}G_{\mathrm{ve}}E_{a}\left[-\frac{{\left(t-s\right)}^{a}G_{\mathrm{ve}}}{b}\right]{\left[\det C\left(t\right)\right]}^{-1/3}P^{4}\left(t\right):\frac{d\bar{e}\left(s\right)}{ds}ds \end{array}.$$

It should be remarked that the commonly used exponential typed relaxation function based on a Maxwell element can be viewed as a special case of the Mittag–Leffler typed relaxation function, which can be illustrated by setting a=1. Subsequently, it is obtained $E_{1}\left(x\right)=\exp\left(x\right)$. Experimental result showed that, normally, the stress relaxation of rubber is very fast at the beginning followed by an extremely slow process. Compared with the exponential typed relaxation function where a large number of parameter are needed to depict the relaxation behavior of rubber material accurately, the merit of the Mittag–Leffler typed relaxation function is that the same level of accuracy can be achieved with less material parameters.

When $G_{\mathrm{ve}}$ is much larger than $G_{\infty }$, the fractional SLS model degrades into the fractional Kelvin-Voigt model [48] and the solution of the stress is
(35)$$\begin{array}{c} S^{\mathrm{ve}}\left(t\right)=G_{\infty }J^{-2/3}\mathrm{Dev}\left(I\right)-b\int_{0}^{t}\frac{{\left(t-s\right)}^{-a}}{\Gamma (1-a)}{\left[\det C\left(t\right)\right]}^{-1/3}P^{4}\left(t\right):\frac{d\bar{e}\left(s\right)}{ds}ds. \end{array}$$

According to the time-temperature superposition principle, at the same frequency, the modulus corresponding to high temperature is much lower than that at low temperatures [68]. According to the research of Kari et al. [49], $G_{\mathrm{ve}}$ can be interpreted as the difference between the shear modulus in the rubbery and glassy regions. Therefore, for the temperatures considered in this paper (242, 247, 254 and 273 K) to model the temperature dependence, $G_{\mathrm{ve}}$ is set to be infinite at 273 K for the parameter identification concern.

Under the time-temperature superposition principle frame, the temperature dependence can be incorporated into the fractional SLS model by introducing a horizontal shift factor $a_{T}$ and a vertical shift factor $b_{T}$ into Equation (Equation 34)
(36)Sve(T,t)=bTSve(t)G∞J−2/3Dev(I)−bT∫0tGveEa−t−st−saTaTaGvebdetCt−1/3P4(t):de¯(s)dsds,
where *T* is an arbitrary temperature. The vertical shift factor $b_{T}$ accounts for the entropic and thermal behavior with
(37)$$b_{T}=\frac{T}{T_{0}}\left[1-\vartheta \left(T-T_{0}\right)\right],$$
where ϑ is the thermal expansion coefficient and $T_{0}$ is the reference temperature. According to Mark [69], $\vartheta \;=\;6.6000\times 10^{-4}\;K^{-1}$ for natural rubber is set for MS rubber since the thermal expansion of iron particles is very small compared with natural rubber. Regarding $a_{T}$, the WLF function
(38)$$a_{T}=10^{-\frac{D_{1}\left(T-T_{0}\right)}{D_{2}+\left(T-T_{0}\right)}},$$
where $D_{1}$ and $D_{2}$ are material parameters is used. For more details of the WLF function, the reader is referred to Ferry [44]. Equations (Equation 35) and (Equation 36) can be numerically implemented through a convolutional approach where both the history of stress and strain are needed [70]. More details regarding fractional integrals can be found in Lubich [71]; Kempfle et al. [72]; Wang and Kari [39]; Alotta et al. [70]; Kari [73,74]. It should be noted that a direct calculation of the Mittag–Leffler function in Equation (Equation 36) is not suggested. As recommended by Haupt and Lion [66], a cumulative relaxation spectrum method to compute the value of the Mittag–Leffler function is more time efficient. According to Haupt and Lion [66], the cumulative spectrum of the Mittag–Leffler function is
(39)$$H\left(v\right)=\frac{1}{a\pi }\left[\left\{\arctan\left[\frac{{\left(v\tau_{a}\right)}^{a}+\cos\left(a\pi \right)}{\sin\left(a\pi \right)}\right]\right\}-\pi \left(\frac{1}{2}-a\right)\right],$$
where
(40)$$\tau_{a}=\frac{b{\left(a_{T}\right)}^{a}}{G_{\mathrm{ve}}}.$$

Then, the value of the Mittag–Leffler function can be calculated by the cumulative spectrum by
(41)Ea−ttaTaTaGveb=∫0∞Hste−sds.

### 3.2. Amplitude Related Temperature Dependence

As illustrated in Figure 1, the amplitude dependence of MS rubber is reflected by a neo-Hookean spring in series with a plastic element. To be specific, a bounding surface nonlinear kinematic hardening model with a zero-elastic range is utilized to serve as the plastic element.

The bounding surface model is developed by Dafalias and Popov [75] with the initial purpose of simulating the mechanical response of artificial graphite under loading. Due to the outstanding performance to describe the plastic response of materials under irregular cyclic and multi-axial loading, the model proposed has been widely used to simulate the plastic behavior of metals and geo-structures [76,77]. Compared with other plastic models, an image stress is assumed to exist in this model and the surface surrounded by the image stress is the bounding surface. The direction of the true stress rate determines the position of the image stress on the bounding surface. Afterwards, the distance between the true stress and the image stress determines the plastic moduli which connects the true stress rate and strain rate.

The constitutive equations for the elastoplastic element are as follows. After utilizing a neo-Hookean quasi-incompressible spring for the elastic part, the stored energy for the elastoplastic element per unit volume is
(42)$$\Omega^{\mathrm{ep}}=\frac{G_{\mathrm{ep}}\left(B_{R}\right)}{2}\left({\bar{I}}_{1}^{e}-3\right)$$
where ${\bar{I}}_{1}^{e}=\mathrm{tr}\left({\bar{C}}^{e}\right)$. Subsequently, according to the Coleman Noll procedure [58], the second Piola-Kirchhoff stress ${\hat{S}}^{\mathrm{ep}}$ in the intermediate configuration is
(43)$${\hat{S}}^{\mathrm{ep}}=2\frac{\partial \Omega^{\mathrm{ep}}}{\partial C^{e}}.$$

By Equation (Equation 10), the Mandel stress $\Sigma^{\mathrm{ep}}$ in the intermediate configuration is
(44)$$\Sigma^{\mathrm{ep}}=2C^{e}\frac{\partial \Omega^{\mathrm{ep}}}{\partial C^{e}}=J^{-2/3}G_{\mathrm{ep}}\mathrm{dev}\left(C^{e}\right),$$
where $\mathrm{dev}\left(\cdot \right)=\left(\cdot \right)-1/3\left[\left(\cdot \right):I\right]I$. A bounding surface is defined in the intermediate configuration through the Mandel stress
(45)$$\Psi =∥\Sigma^{\mathrm{image}}-\beta ∥-S_{\;\mathrm{bounding}}=0,$$
where $∥\cdot ∥$ denotes the Frobenius norm, $\Sigma^{\mathrm{image}}$ is the corresponding image Mandel stress on the bounding surface, β is the center, also referred to as the back stress of the bounding surface, and $S_{\;\mathrm{bounding}}$ is the radius of the bounding surface. Mróz kinematic rule [78] is used to determine the image stress $\Sigma^{\mathrm{image}}$. The idea is that $\Sigma^{\mathrm{image}}$ is positioned as the intersection point of the extension of ${\dot{\Sigma }}^{\mathrm{ep}}$ with the bounding surface. Therefore,
(46)$$\Sigma^{\mathrm{image}}\;=\;\Sigma^{\mathrm{ep}}\;+\;\delta m,$$
where m=Σ˙epΣ˙epΣ˙epΣ˙ep is the increment direction of $\Sigma^{\mathrm{ep}}$ and δ is the distance from $\Sigma^{\mathrm{ep}}$ to the bounding surface. After determining $\Sigma^{\mathrm{image}}$, the normal direction of the bounding surface
(47)$$n=\frac{\partial \Psi }{\partial \Sigma^{\mathrm{image}}}=\frac{\Sigma^{\mathrm{image}}-\beta }{∥\Sigma^{\mathrm{image}}-\beta ∥}$$
is obtained. An associated plasticity assumption [79] where the direction of the plastic deformation gradient rate $D_{p}$ is the same compared with the normal direction n is postulated. Therefore,
(48)$$D_{p}={\dot{F}}^{p}{\left(F^{p}\right)}^{-1}\;=\;\lambda n,$$
where λ is the magnitude of $D_{p}$ named as plastic multiplier and the maximum dissipation principle of the elastoplastic constitutive branch can be guaranteed. According to Österlöf et al. [80], the hardening rule is
(49)$${\dot{\Sigma }}^{\mathrm{ep}}:n=H\lambda ,$$
where
(50)$$H=H_{p}\left(\frac{\delta_{\mathrm{in}}}{\delta_{\mathrm{in}}-\delta }\right)$$
is the plastic modulus connecting the stress rate and the strain rate and $\delta_{\mathrm{in}}$ is the discrete memory parameter which remembers the largest distance from the elastoplastic stress to the corresponding image stress since the last turning point. The evolution law of β is
(51)$$\dot{\beta }=\frac{H_{p}}{H}{\dot{\Sigma }}^{\mathrm{ep}}.$$

Concerning the temperature dependence of the elastoplastic part, follwing the pattern of Muhr [55], the Arrhenius function is utilized. The relationship between the elastoplastic stress between the reference temperature $T_{0}$ and an arbitrary temperature *T* is
(52)$$\Sigma^{\mathrm{ep}}\left(T\right)\;=\;\Sigma^{\mathrm{ep}}\left(T_{0}\right)e^{\left(\frac{E}{RT}-\frac{E}{RT_{0}}\right)},$$
where *E* is a material parameter and R=8.31 J ${\mathrm{mol}}^{-1}$
$K^{-1}$ is the Boltzmann constant. The numerical implementation method related to the elastoplastic branch is introduced in Appendix A.

### 3.3. Magnetic Dependence of MS Rubber

The observed magnetic dependence of the modulus of MS rubber is a contribution of two kind of physical mechanisms. The first one is related to the increase of internal energy caused by the application of magnetic field. The equations to describe this mechanism have already been introduced in Equations (Equation 23) and (Equation 26).

For the second mechanism, according to Jolly et al. [21], two adjacent particles have opposite magnetic charges at the position where they lie close to each other along the direction of the applied magnetic field after magnetization. Due to the magnetic dipole-dipole interaction, there is an enhancement in the modulus macroscopically. Dynamic shear modulus measurement conducted by Blom and Kari [25] revealed that there is a strong correlation between magnetic dependence and amplitude dependence. To be specific, a smaller strain amplitude leads to a larger magnetic dependence. Therefore, $G_{\mathrm{ep}}$, $H_{p}$ and $S_{\mathrm{bounding}}$ are set to be a function of an applied magnetic flux density to represent the correlation between the magnetic dependence and amplitude dependence. In addition, a magnetic dependence is introduced for $G_{\infty }$ as well. Furthermore, measurement results show that the shear modulus of MS rubber increases with the increasing magnetic field until magnetic saturation is reached. To describe the magnetic enhancement and magnetic saturation of the modulus, similar to Saxena et al. [38], a hyperbolic tangent typed function is used for $G_{\mathrm{ep}}$, $H_{p}$, $G_{\infty }$ and $S_{\;\mathrm{bounding}}$ by
(53)$$G_{\mathrm{ep}}=G_{\mathrm{ep}0}\left[1+\kappa_{1}\tanh\left(\frac{I_{4}}{I_{s}}\right)\right],$$
(54)$$H_{p}\;=\;H_{p0}\left[1+\kappa_{2}\tanh\left(\frac{I_{4}}{I_{s}}\right)\right],$$
(55)$$G_{\infty }\;=\;G_{\infty 0}\left[1+\kappa_{3}\tanh\left(\frac{I_{4}}{I_{s}}\right)\right]$$
and
(56)$$S_{\;\mathrm{bounding}}\;=\;S_{\mathrm{bounding}0}\left[1+\kappa_{4}\tanh\left(\frac{I_{4}}{I_{s}}\right)\right],$$
where $I_{4}=B_{R}\cdot B_{R}$ is the scalar invariant for the magnetic flux density $B_{R}$ inside MS rubber in the reference configuration and $I_{s}$ is used to represent the magnetic saturation of MS rubber, which has the same units as $I_{4}$. Parameters $G_{\mathrm{ep}0}$, $H_{p0}$, $G_{\infty 0}$ and $S_{\mathrm{bounding}0}$ are material parameters at zero magnetic field. Symbols $\kappa_{1}$, $\kappa_{2}$, $\kappa_{3}$ and $\kappa_{4}$ are positive real values to reflect the magnetic enhancement. It should be noted that, normally, only the magnetic flux density outside MS rubber is known. The determination of the magnetic flux density inside MS rubber can be achieved by Equations (Equation 11) to (Equation 13) along with Equation (Equation 21) through a semi-inverse method. The details are explained in Appendix B.

## 4. Results and Discussion

The dynamic shear modulus measurement results of isotropic MS rubber under four kinds of magnetic field (0, 0.33, 0.55 and 0.8 T), three kinds of strain amplitude (0.0005, 0.0015 and 0.005) with discrete temperatures (242, 247, 249 and 273 K) ranging from 200 to 900 Hz conducted by Lejon and Kari [41] were utilized for the parameter identification of the developed model. The deformation gradient is
(57)$$F=\left[\begin{array}{ccc} 1 & \gamma & 0\\ 0 & 1 & 0\\ 0 & 0 & 1 \end{array}\right],$$
where $\gamma \;=\;A\sin\left(\omega t\right)$ with *A* the strain amplitude and ω the angular frequency. Regarding the method of parameter identification, the nonlinear least square fitting method and the *lsqnonlin* corresponding command in Matlab^®^ (MATLAB Release 2015b, The MathWorks, Inc., Natick, MA, USA) is utilized.

### 4.1. Simulation of the Magnetic, Frequency and Amplitude Dependence

Firstly, measurement data at 273 K (reference temperature) and 0 T with strain amplitudes 0.0005, 0.0015 and 0.005 ranging from 200 to 900 Hz were utilized to identify the basic material parameters $G_{\infty 0}$, *a*, *b*, $H_{p0}$, $S_{\mathrm{bounding}0}$ and $G_{\;\mathrm{ep}0}$ with $G_{\;\mathrm{ve}}$ is set to be infinite. For this round of parameter identification, the Mittag–Leffler function is not used since when $G_{\;\mathrm{ve}}$ is infinite, Equation (Equation 34) is degraded to Equation (Equation 35). For each set of material parameters, the corresponding stress-strain response can be obtained. While there is a nonliner-viscoelastic behavior (strain dependence) of MS rubber, the commonly used viscoelastic equivalent method [55] and the Fourier transformation appoarch is adopted in this paper to extract the dynamic shear modulus of MS rubber. The total stress obtained from the constitutive model is in Equation (Equation 24) with shear stress $\sigma_{12}$. Then, the equivalent modelled dynamic shear modulus is
(58)$$\tilde{G^{*}}=\frac{\tilde{\sigma_{12}}}{\tilde{\gamma }},$$
where the wavy line superscript represents the Fourier transform operator. Subsequently, the norm of the difference between $\tilde{G^{*}}$ and the measured dynamic modulus is set as the objective function. By using the function lsqnonlin in MATLAB^®^ (MATLAB Release 2015b, The MathWorks, Inc., Natick, MA, USA), the objective function can be minimized. After several times of iteration, the identified parameters are $G_{\infty 0}=6.7584\times 10^{6}\;N\;m^{-2}$, a=0.4368, $b=1.4077\times 10^{5}\;N\;s^{a}\;m^{-2}$, $H_{p0}=0.7828\times 10^{6}\;N\;m^{-2}$, $S_{\;\mathrm{bounding}0}=7.3780\times 10^{3}\;N\;m^{-2}$ and $G_{\mathrm{ep}0}=3.4095\times 10^{6}\;N\;m^{-2}$. The relative difference between the measurement and simulation result is 4.50%. After the first round of parameter identification, a second round is conducted to obtain the magnetic related parameters $\kappa_{1}$, $\kappa_{2}$, $\kappa_{3}$, $\kappa_{4}$ and $I_{s}$. With a relative difference of 5.60%, the identified parameters are $\kappa_{1}\;=\;1.6271$, $\kappa_{2}\;=\;0$, $\kappa_{3}\;=\;2.2815$, $\kappa_{4}\;=\;0.0268$ and $I_{s}={\left(0.3309\;T\right)}^{2}$. By applying the Fourier transformation method as in Blom and Kari [30], the result of the dynamic modulus can be obtained. The comparison between the simulation and measurement results are shown in Figure 6, Figure 7 and Figure 8. It can be found that by using the developed model, the magnetic, frequency and amplitude dependence of MS rubber can be well depicted. It should be noted that the bump for the magnitude and loss factor between 600 and 800 Hz is related to the resonance of the test machine. Regarding the loss factor, visibly, the main trend of the measurement loss factor is that they overlay with each other at different magnetic fields. Yet, the local waviness of the measured result indicates the measurement error and the difficulty to determine the exact value of loss factor experimentally. Even though the loss factor is slightly underestimated, the main trend that the loss factor under different magnetic fields follows the main curve is well depicted by the model. Therefore, the fitting ability of the loss factor is satisfactory.

To check the ability of the developed model to capture the magnetic saturation, the comparison of the dynamic shear modulus magnitude at 400 and 700 Hz, respectively, with three kinds of strain amplitude at different magnetic fields is shown in Figure 9. It can be found that magnetic dependence is slightly overestimated by the model developed. However, the magnetic saturation of the dynamics shear modulus can be replicated properly by the developed model.

### 4.2. Simulation of the Temperature Dependence

After identifying the basic material parameters at 273K, another round of parameter identification is conducted. Equations (Equation 36)–(Equation 41) are used. The temperature-dependent material parameters are $G_{\mathrm{ve}}=1.9630\times 10^{10}\;N\;m^{-2}$, $D_{1}=5.6154$, $D_{2}=89.2400\;K$ and $E=2.7000\times 10^{4}\;J\;{\mathrm{mol}}^{-1}$.

The dynamic shear modulus magnitude and loss factor under different temperatures at different strain amplitudes under 0 and 0.55 T, respectively are shown in Figure 10, Figure 11, Figure 12, Figure 13, Figure 14 and Figure 15. Measurement results revealed that MS rubber has already reaches magnetic saturation at 0.55 T, therefore, comparison between measurement and simulation data at 0.8 T is not shown. Furthermore, the data at 0.3 T lies between the result at 0 and 0.5 T. To keep it compact, only results at 0 and 0.55 T are displayed. Generally speaking, the trend that the magnitude of the dynamic shear modulus increases with decreasing temperature is well captured by the developed model. A possible explanation for the deviation of the magnitude of the dynamic shear modulus at low temperatures could be that MS rubber may have already reached the glassy region and partially crystallized. The WLF function is applicable when the temperatures are around the transition temperature, therefore, a manual shifting of $a_{T}$ may be needed at lower temperatures. The wide fluctuation of the measured loss factor throughout all zones indicates the difficulty to determine the loss factor experimentally. This provides a possible explanation for the less satisfactory fitting of the loss factor by the model.

## 5. Conclusions

A constitutive model based on continuum mechanics with the Helmholtz free energy assumption and magnetic theory is developed. The constitutive model consists of a fractional SLS element, a bounding surface-based elastoplastic element and a magnetic stress tensor term in parallel. A hyperbolic typed tangent function is utilized to reflect the magnetic enhancement of MS rubber after applying of the magnetic field. After developing the model, the material parameters of the constitutive equations are identified by using the nonlinear least square method. A comparison between the measurement and simulation results of the magnitude and loss factor for the dynamic shear modulus of MS rubber indicates that with few parameters, the amplitude, frequency and magnetic dependence of MS rubber can be well reflected by the proposed constitutive model.

Furthermore, by introducing the WLF function and the Arrhenius function into the fractional SLS model and elastoplastic element, respectively, the developed model is augmented and can be used to predict the mechanical response of MS rubber under different temperatures. The simulation results reveal that the fitting of the dynamic shear modulus magnitude of MS rubber with temperatures around the glassy transition temperature by the model is good. However, the prediction of the loss factor of the dynamic shear modulus by the constitutive model and the dynamic shear modulus at low temperatures needed to be further improved. Furthermore, it should be remarked that the model developed in this paper is not a fully coupled thermo-mechanical constitutive model. The stress caused by material expansion due to the temperature increase and the increase of temperature due to the movement of the material is outside of the scope of this paper.

In summary, the constitutive model developed herein to reflect the amplitude, frequency, magnetic and temperature-dependent mechanical properties is very important for the prediction of MS rubber-based anti-vibration devices mechanical performance in the design phase and helpful for the estimation of the mechanical performance and reliability of MS rubber-based devices at different magnetic fields, frequencies, strain amplitudes and temperatures.

## Figures and Tables

**Figure 1 polymers-13-00472-f001:**
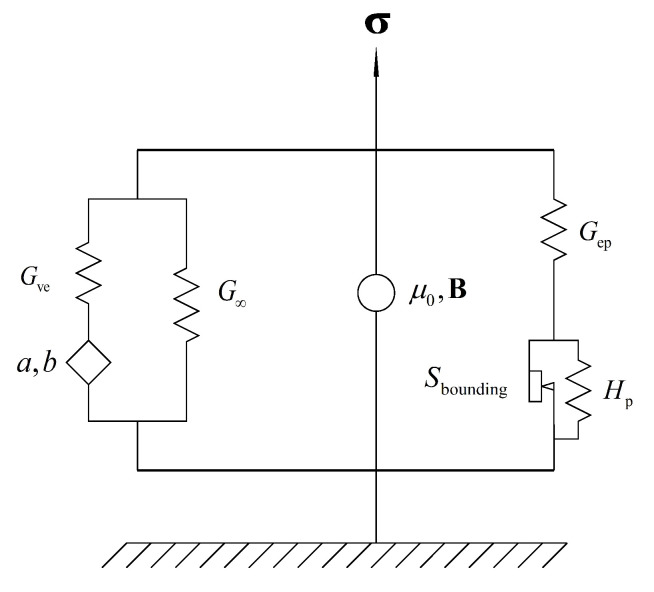
Rheological illustration of the constitutive model.

**Figure 2 polymers-13-00472-f002:**
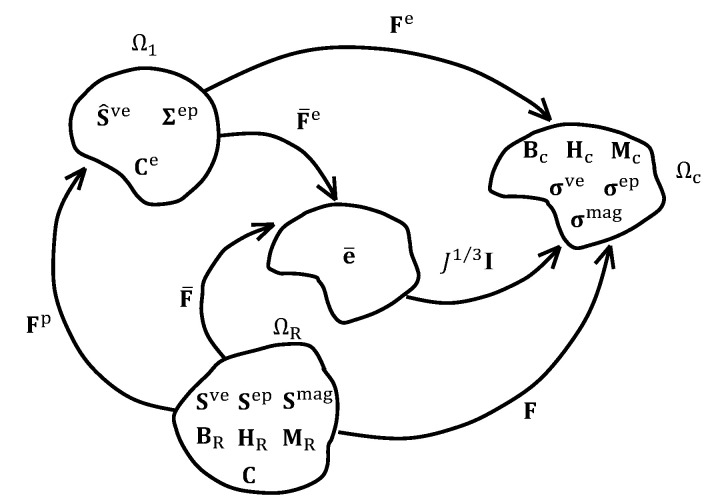
The reference, intermediate, volume preserving and current configurations and the corresponding tensors.

**Figure 3 polymers-13-00472-f003:**
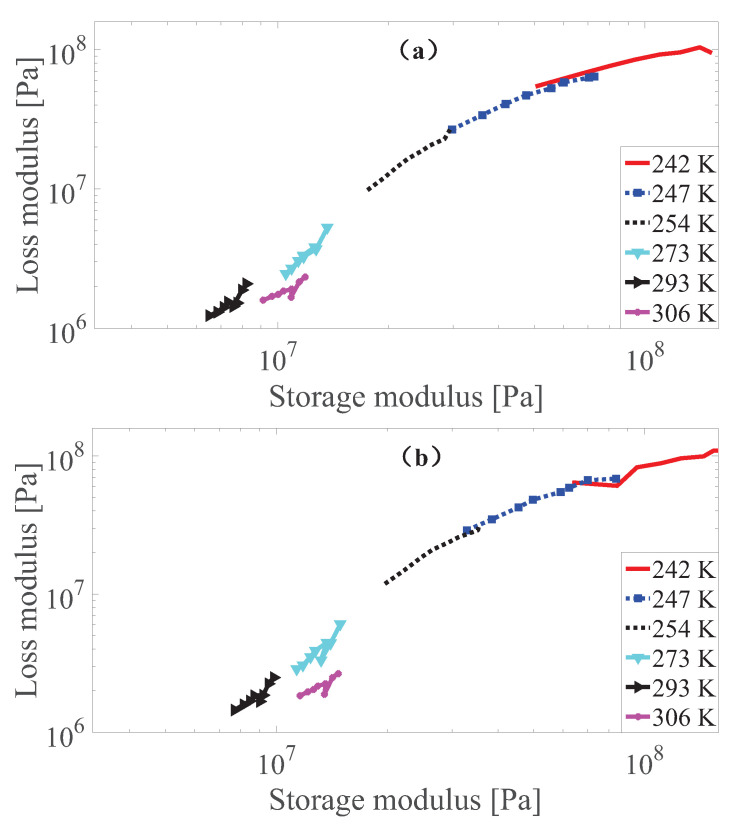
The Cole–Cole plot of dynamic shear modulus at 0.0005 strain amplitude with 0 T (**a**) and 0.55 T (**b**) under different temperatures.

**Figure 4 polymers-13-00472-f004:**
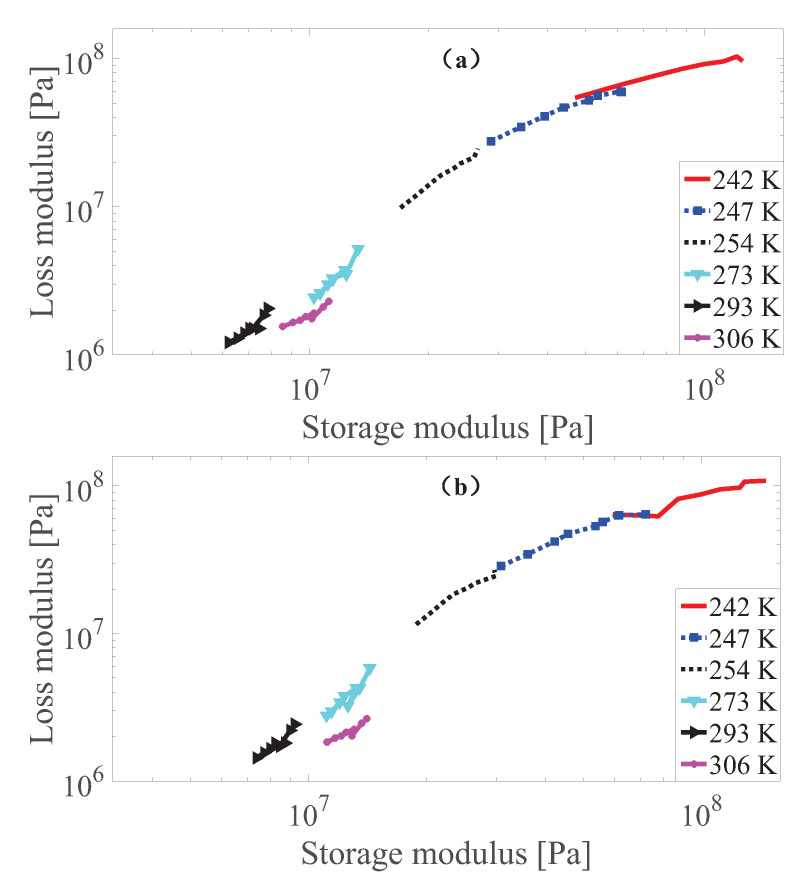
The Cole–Cole plot of dynamic shear modulus at 0.0015 strain amplitude with 0 T (**a**) and 0.55 T (**b**) under different temperatures.

**Figure 5 polymers-13-00472-f005:**
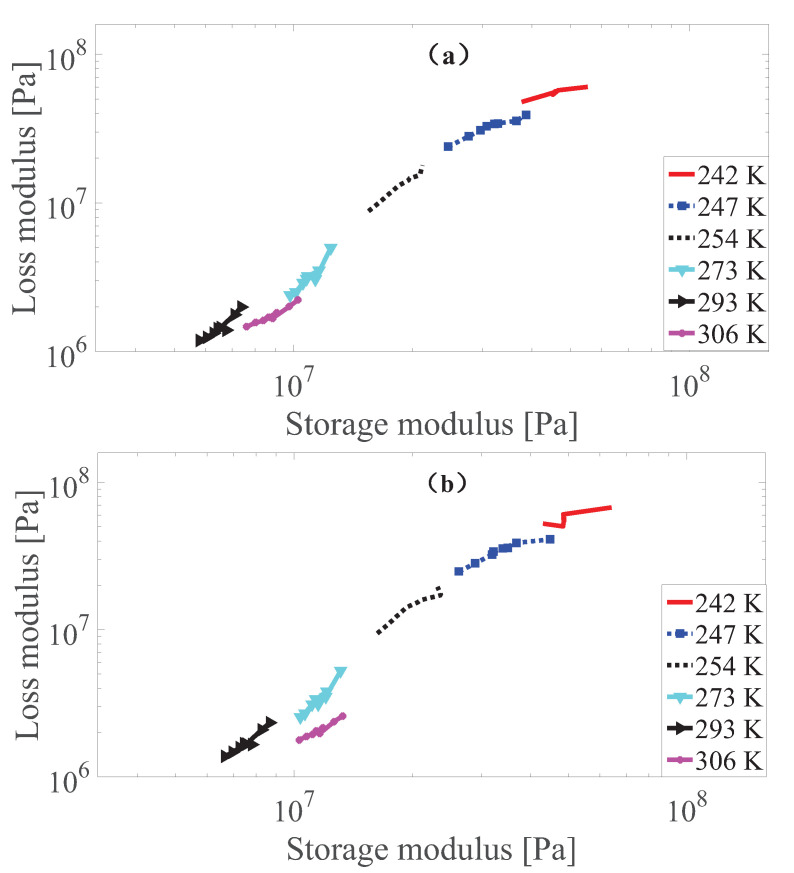
The Cole–Cole plot of dynamic shear modulus at 0.005 strain amplitude with 0 T (**a**) and 0.55 T (**b**) under different temperatures.

**Figure 6 polymers-13-00472-f006:**
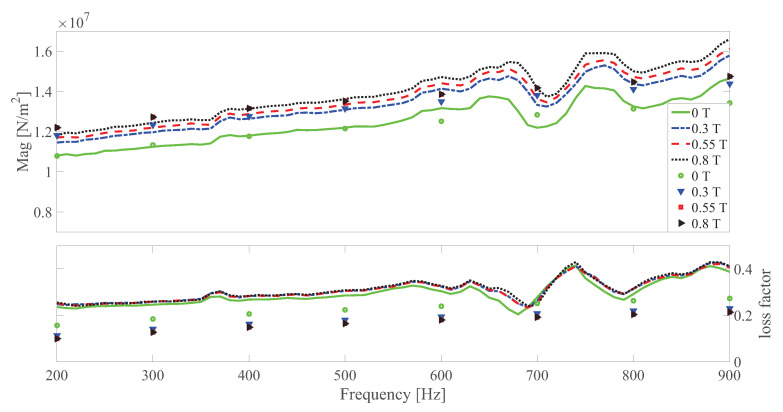
The magnitude and loss factor of the dynamic shear modulus versus frequency at 0.0005 strain amplitude and 273 K. Lines and symbols are the measurement and simulation results, respectively.

**Figure 7 polymers-13-00472-f007:**
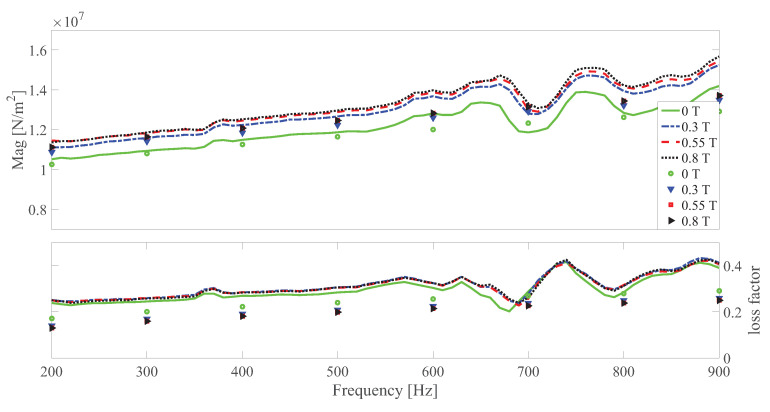
The magnitude and loss factor of the dynamic shear modulus versus frequency at 0.0015 strain amplitude and 273 K. Lines and symbols are the measurement and simulation results, respectively.

**Figure 8 polymers-13-00472-f008:**
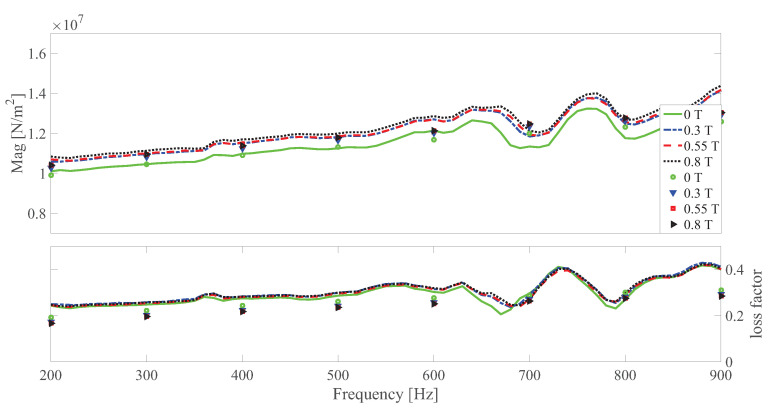
The magnitude and loss factor of the dynamic shear modulus versus frequency at 0.005 strain amplitude and 273 K. Lines and symbols are the measurement and simulation results, respectively.

**Figure 9 polymers-13-00472-f009:**
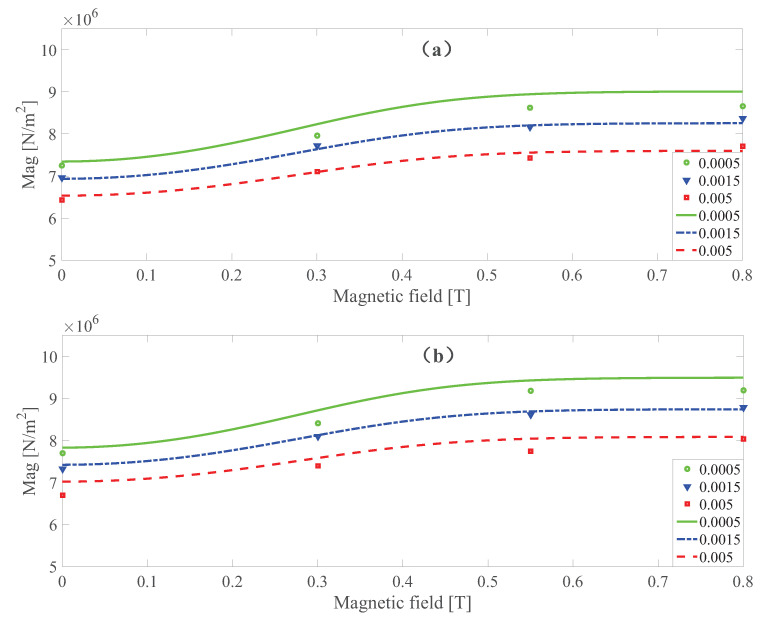
The magnitude of the dynamic shear modulus versus magnetic field at 400 Hz (**a**) and 700 Hz (**b**), respectively, under different strain amplitudes at 273 K. Lines and symbols are the simulation and measurement results, respectively.

**Figure 10 polymers-13-00472-f010:**
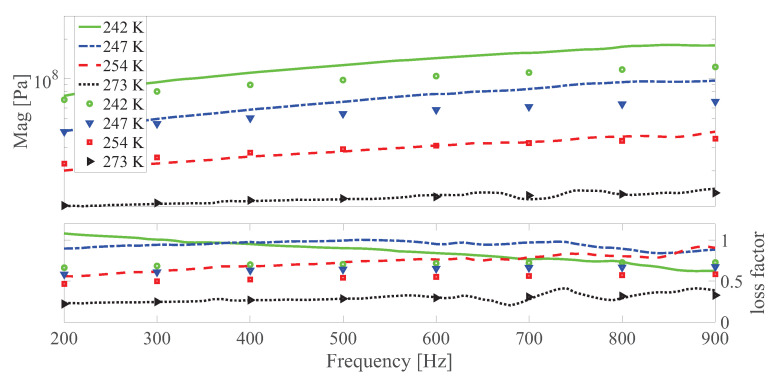
The magnitude and loss factor of the dynamic shear modulus versus frequency at 0.0005 strain amplitude and 0 T under different temperatures. Lines and symbols are the simulation and measurement results, respectively.

**Figure 11 polymers-13-00472-f011:**
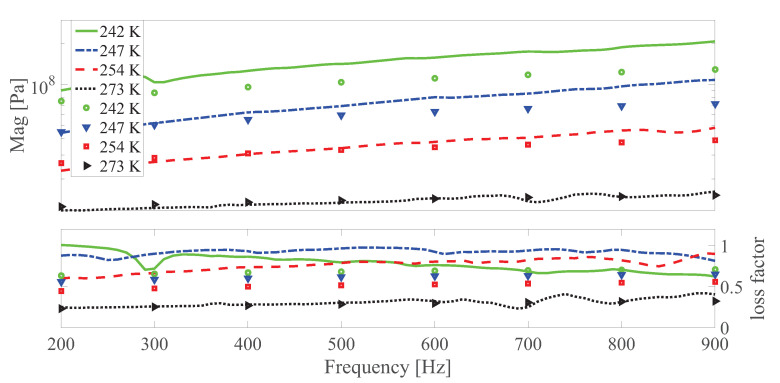
The magnitude and loss factor of the dynamic shear modulus versus frequency at 0.0005 strain amplitude and 0.55 T magnetic field under different temperatures. Lines and symbols are the simulation and measurement results, respectively.

**Figure 12 polymers-13-00472-f012:**
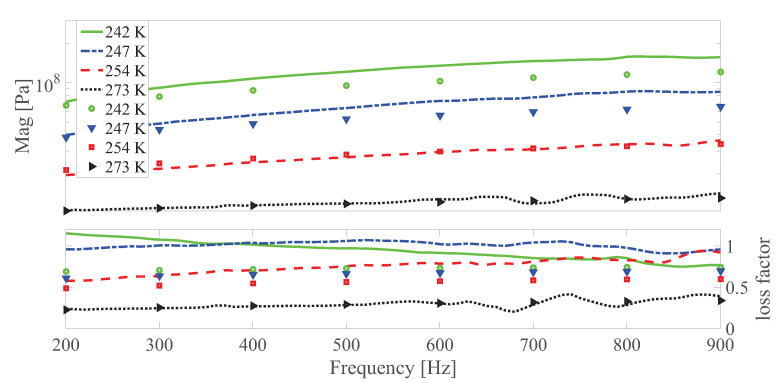
The magnitude and loss factor of the dynamic shear modulus versus frequency at 0.0015 strain amplitude and 0 T magnetic field under different temperatures. Lines and symbols are the simulation and measurement results, respectively.

**Figure 13 polymers-13-00472-f013:**
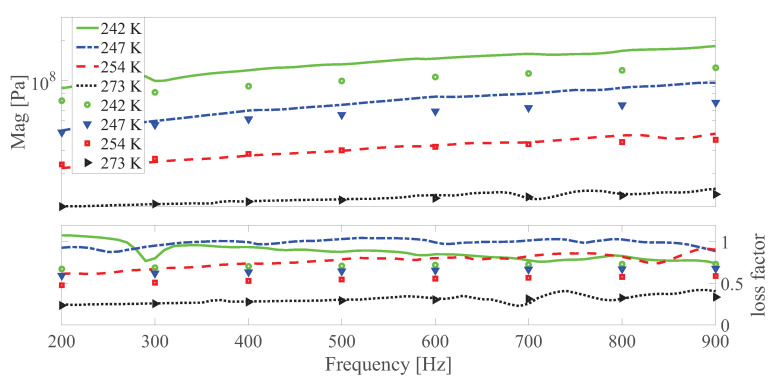
The magnitude and loss factor of the dynamic shear modulus versus frequency at 0.0015 strain amplitude and 0.55 T under different temperatures. Lines and symbols are the simulation and measurement results, respectively.

**Figure 14 polymers-13-00472-f014:**
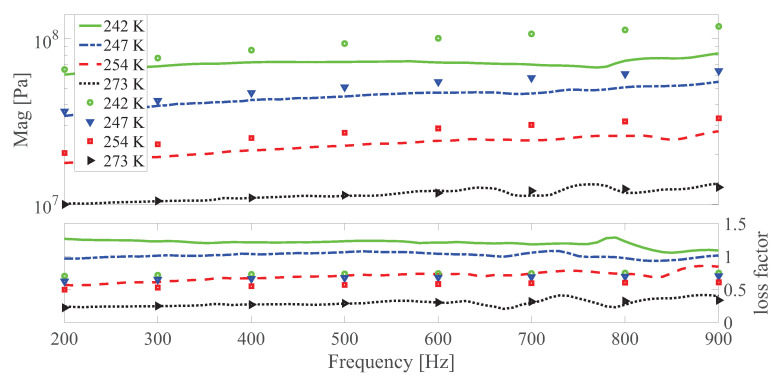
The magnitude and loss factor of the dynamic shear modulus versus frequency at 0.005 strain amplitude and 0 T magnetic field under different temperatures. Lines and symbols are the simulation and measurement results, respectively.

**Figure 15 polymers-13-00472-f015:**
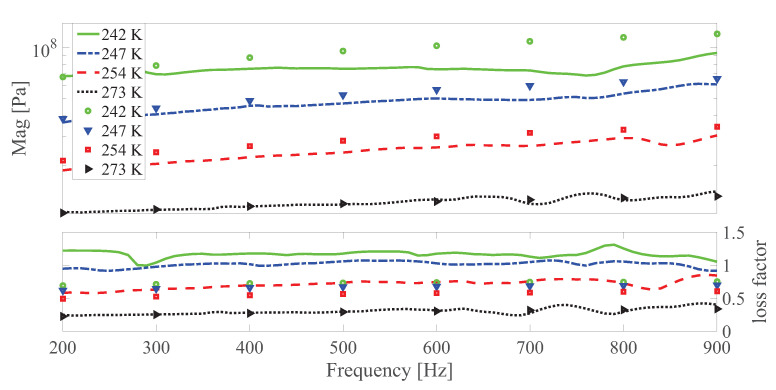
The magnitude and loss factor of the dynamic shear modulus versus frequency at 0.005 strain amplitude and 0.55 T magnetic field under different temperatures. Lines and symbols are the simulation and measurement results, respectively.

**Table 1 polymers-13-00472-t001:** Composition of magneto-sensitive (MS) rubber.

Substance	Parts per Hundred Rubber
Natural rubber	100
Zinc oxide	6
Stearine	0.5
Sulphur	3.5
Mercaptobenzothiazole	0.5
Hydrocarbon oil	40
Nytex 480 plasticizer	40

## Data Availability

The data presented in this study are available on request from the corresponding author. The data are not publicly available due to the possible relevance between the subsequent research and the data used in this article.

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
