# Peer review of "Constitutive Model of Isotropic Magneto-Sensitive Rubber with Amplitude, Frequency, Magnetic and Temperature Dependence under a Continuum Mechanics Basis"

_polymers, 2021, doi:10.3390/polym13030472_

Round 1

Reviewer 1 Report

The manuscript is well and timely written. However, there are some issues which the Authors need to take into account before the article can be accepted for publication:

1. Besides enumerating many possible applications and constitutive models, the Introduction part shall include a short description of other approaches used in the literature for similar or materials. For example, in the papers:

https://doi.org/10.1016/j.jiec.2013.12.102

and

https://doi.org/10.1016/j.jiec.2017.07.039   the Authors present the fabrication of hybrid magnetorheological elastomers in which besides the magnetic phase, there were introduced graphene nanoparticles. Further, they investigate the influence of an extenal magnetic field and of a compression pressure on electrical properties. Theoretical models which explain the obtained results are developed by using elements of elasticity theory.   2. Figure 2 and A1 are hardly readable. Please improve their quality.   3. Figures 3, 4, 5 and 9: the labels (a) and (b) are too far with respect to the graph. Also the text label shall be slightly smaller (with fonts of similar size as in the main text).   4. What is the reason you choose 242, 247, 254, 273, 293 and 306 K for the temperature?   5. Line 294: "maxwell" shall be capitalized.

Author Response

We would like to thank the reviewer for the careful and thorough reading of our manuscript. At the same time, the thoughtful and constructive suggestions, which are helpful for the improvement of the quality of the manuscript, are also appreciated. Our responses are as follows.

1. Besides enumerating many possible applications and constitutive models, the introduction part shall include a short description of other approaches used in the literature for similar of materials. For example in the papers:https://doi.org/10.1016/j.jiec.2013.12.102 and https://doi.org/10.1016/j.jiec.2017.07.039. The authors present the fabrication of hybrid magnetorheological elastomer in which besides the magnetic phase, there were introduced graphene nanoparticles. Furthermore, they investigate the influence of an external magnetic field and of a compression pressure on electrical properties. Theoretical models which explain the obtained results are developed by using element of elasticity theory.

Reply: We appreciate the comments of the reviewer regarding the literature review for the application of magnetorheological elastomer used in the electrical sensing area. The papers recommended by the reviewer extend our understanding of this magic material. Therefore, the new introduction related to the recommended paper is as follow:

Furthermore, by adding electrical conductive materials in MS rubber to increase the conductivity of MS rubber, an electrical sensitivity of MS rubber will exhibit in addition to the magneto sensitivity. For example, Bica et al. [15,16] fabricated a magnetoresistive sensor by adding graphene nanoparticles in MS rubber and then a theoretical model was proposed to explain the effect of the magnetic field intensity and pressure on the electrical conductivity of MS rubber. Similar research can be found in Wang et al [17], Yun et al. [18], Hu et al. [19], Ding et al. [20]. Due to the electromagnetic mechanical coupling effect, a great potential is shown for MS rubber in the area of vibration control, micro-electro-mechanical system and intelligent sensing and actuating.

2. Figure 2 and A1 are hardly readable. Please improve their quality.

Reply: We appreciated the editorial suggestion regarding the figures of the paper from the reviewer. For the revised version, Figures 2 and A1 with a higher resolution are updated.

3. Figure 3,4,5 and 9, the labels (a) and (b) are too far with respect to the graph. Also the text label shall be slightly smaller with fonts of similar size as in the main text.

Reply: We appreciated the editorial suggestion from the reviewer. For the latest version, the position of the labels are adjusted to the right place. Besides, the text labels for all the figures are adjusted to make sure that the fonts size are similar to the main text.

4. What is the reason you choose 242, 247, 254, 273, 293 and 306K for the temperature.

Reply: Actually, the measurement was conducted about seven years ago. In order to control the temperature, the MS rubber sample and the coil were enclosed in a thermally insulated box. Above room temperatures, hot water was send into the pipe by a water pump to control the temperature in the box, Below room temperatures, liquid nitrogen was released in the box to control the temperatures. The experimental conditions was quite limited at that time, therefore, the measurement temperatures are not consecutive. Regarding the range of temperature. Since we would like to find the change of the mechanical properties of MS rubber from glassy to highly elastic region, the temperatures from 242 to 306 K were selected.

5. Line 294: Maxwell shall be capitalized.

Reply: We appreciated the suggestion from the editor regarding the typos in our manuscript. For the latest version, the typos Maxwell is revised to Maxwell.

Lastly, thanks again for the time and patience spend as well as the constructive suggestions which really help for our research.

Best regards.

Reviewer 2 Report

Good work. The following points are for consideration in minor revision.

  • Proof reading is required. There are typos. E.g., it is assumed that rubber material is thermorheologica simple.
  • As a composite, during cyclic loading of the magnetic field, in particular at very high frequency, energy dissipation may induce heat and thus, the temperature of the composite increases. Can this be included in this model?
  • Size effect, which includes the influence of non-uniformity of the applied magnetic field, and thermal gradient (due to self-heating).
  • 6-9. Rescale the vertical axis to zoom into the range of the results for a clear view.

Author Response

We would like to thank the reviewer for the careful and thorough reading of our manuscript. At the same time, the thoughtful and constructive suggestions, which are helpful for the improvement of the quality of the manuscript, are also appreciated. Our response are as follows.

1. Proof reading is required. There are typos. E.g. It is assumed that rubber material is thermorheologica simple.

Reply: We appreciated the editorial suggestion regarding the typos of the paper from the reviewer. For the revised version, proofreading is done and the mistakes regarding the typos are all amended.

2. As a composite, during cyclic loading of the magnetic field, in particular at very high frequency, energy dissipation may induce heat and thus, the temperature of the composite increases. Can this be included in this model?

Reply: We agree with the reviewer that energy dissipation may induce heat and cause the increase of temperature of MS rubber. The main research of our manuscript is to model the magnetic, frequency, amplitude and temperature dependence of MS rubber under isothermal process. As mentioned in the conclusion section “It should be remarked that the model developed in this paper is not a fully coupled thermo-mechanical constitutive model. The stress caused by material expansion due to the temperature increase and the increase of temperature due to the movement of the material is outside of the scope of this paper”. However, for application concern, the effect of temperature increase by the vibrating of MS rubber is important. We will research on this topic in the follow up.

3. Size effect, which includes the influence of non-uniformity of the applied magnetic field and thermal gradient due to self heating.

Reply: We appreciated the comment from the reviewer regarding the inhomogeneity of the microscopic level of MS rubber due to thermal gradient and the difference magnetic permeability between magnetic particles and rubber material. In our manuscript, a continuum mechanical approach with the basic assumption that the material is viewed as a continuous medium is used. From the constitutive modeling level, we assumed that the temperature in the representative volume unit is constant. After developing the corresponding finite element model regarding the constitutive model and introduce the boundary conditions, the effect of thermal gradient can be considered. However, the topic is out of scope for the current manuscript and will be discussed in the following research. Regarding the effect of non-uniformity, from macroscopic level, the behavior of the representative volume unit with many particles is our main concern instead of a specific particle. Besides, according to the research by D. Mukherjee https://doi.org/10.1016/j.ijnonlinmec.2019.103380, it can be found that on the condition that for small to moderate particle volume fractions, the analytical homogenization continuum mechanical based constitutive model is sufficient to describe the mechanical performance of MS rubber. Therefore, currently, the magnetic field non-uniformity is not considered in our manuscript.

4. Rescale the vertical axis in figures 6-9 to zoom into the range of the results for a clear view.

Reply: We appreciate the comment from the reviewer regarding the quality improvement of the figures in our manuscript. The vertical axis in figures 6-9 are zoomed into the right range for the latest version of the manuscript.

Lastly, thanks again for the time and patience spend as well as the constructive suggestions, which really helps for our research.

Best regards.
